# Natural Variations in Key Maturity Genes Underpin Soybean Cultivars Adaptation Beyond 50° N in Northeast China

**DOI:** 10.3390/ijms26073362

**Published:** 2025-04-03

**Authors:** Hongchang Jia, Baiquan Sun, Bingjun Jiang, Peiguo Wang, Mahmoud Naser, Shuqing Qian, Liwei Wang, Lixin Zhang, Mikhail Sinegovskii, Shi Sun, Wencheng Lu, Valentina Sinegovskaya, Jiangping Bai, Tianfu Han

**Affiliations:** 1State Key Laboratory of Aridland Crop Science/College of Agronomy, Gansu Agricultural University, Lanzhou 730070, China; jiahongchang@haas.cn (H.J.); wpg15101224039@163.com (P.W.); 2Institute of Crop Sciences, Chinese Academy of Agricultural Sciences, Beijing 100081, China; sunbaiquan@caas.cn (B.S.); jiangbingjun@caas.cn (B.J.); m.abdelmoteleb@agr.dmu.edu.eg (M.N.); xaqsq13@126.com (S.Q.); wangliwei02@caas.cn (L.W.); zhanglixin_1994@163.com (L.Z.); sunshi@caas.cn (S.S.); 3Heihe Branch, Heilongjiang Academy of Agricultural Sciences, Heihe 164399, China; luwencheng@haas.cn; 4National Nanfan Research Institute, Chinese Academy of Agricultural Sciences, Sanya 572000, China; 5Far East Research Institute of Agriculture, Khabarovsk Federal Research Center of the Far Eastern Branch of the Russian Academy of Sciences, Khabarovsk 680521, Russia; sinmikhail@gmail.com (M.S.); valsin09@gmail.com (V.S.)

**Keywords:** soybean, maturity group, photothermal adaptation, high-latitude cultivation, adaptability, haplotype

## Abstract

Expanding soybean planting is vital for food security both in China and globally. The 50° N latitude serves as the northern boundary of major soybean regions. However, enhancing the adaptability of soybean to photothermal conditions enables the potential to extend cultivation to higher latitudes and altitudes. Understanding the genetic basis of super-early maturity of soybean is crucial to achieving this goal. In this study, 438 soybean germplasms collected from high-latitude regions were evaluated in Heihe (HH) (50°15′ N, 127°28′ E, 154 m), Beijicun (BJC) (53°28′ N, 122°21′ E, 295 m) and Labudalin (LBDL) (50°15′ N, 120°19′ E, 577 m). Using resequencing data, we analyzed natural variation and haplotypes in 35 key genes associated with flowering time and maturity. The results showed that the relative maturity groups (RMGs) for BJC, HH, and LBDL were −1.0, 0.0, and −1.2, respectively. Among the 35 genes analyzed, 23 had identical allelic variations, while 12 genes exhibited 19 SNPs and four InDels. Functional mutations were identified in *E1*, *E2*, *E3*, and *E4*. Notably, all cultivars carried the *e1-as* allele of *E1*, which is likely critical for high-latitude adaptation. Additional mutations included a single-base substitution in *E2* (16142 A > T) and *E3* (5203 C > T), causing premature codon termination, along with frameshift mutations in *E4* (3726 and 4099) and *E3* (2649). Haplotype analysis revealed significant differences in growth stages among nine gene haplotypes. The higher frequency of early-maturing haplotypes in BJC and LBDL highlights the role of gene accumulation in soybean adaptation. These findings offer valuable insights for improving soybean maturity and expanding its cultivation in high-latitude regions of China.

## 1. Introduction

The soybean (*Glycine max*), domesticated from its wild progenitor *Glycine soja* in the temperate regions of China some 6000 to 9000 years ago, has become a fundamental component of global agriculture [1,2]. The principal site of domestication is thought to be the Huang-Huai-Hai region of China, located between latitudes 32° N and 40° N [3]. Throughout the years, soybean agriculture has expanded outside its native region, currently spanning latitudes from 53° N to 35° S [3].

Currently, soybean serves as an essential source of plant protein and edible oil, while also significantly contributing to global food security [3]. Although China is still one of the top five soybean producers globally [4], it imports more than 80% of its annual soybean needs, absorbing over 120 million tons, whilst domestic production has been restricted to merely 20 million tons in recent years [5]. This significant dependence on imports is a fundamental vulnerability for the country’s food and energy security [6].

Northeast China is a crucial soybean-producing area, accounting for nearly two-thirds of the national output and ensuring a stable domestic supply [7]. Environmental factors, including light and temperature, restrict soybean farming in the northernmost areas of the region, with the 50th parallel north delineating the northern boundary of the primary production zone [8]. Extending soybean farming beyond the 50th parallel and west of the Greater Khingan Mountains is essential for enhancing production in high-latitude and high-altitude areas, thus bolstering China’s self-sufficiency in soybean production [9].

Soybean is a typical warm-season, short-day crop. Expanding soybean cultivation to higher latitudes and altitudes requires improving cultivar adaptability to longer photoperiods and lower temperatures [10,11,12]. Therefore, understanding the genetic basis of super-early flowering and maturity of soybean is crucial for improving its adaptation to stress-stricken long-day and low-temperature environments [13,14].

The photoperiodic regulation of flowering in soybean largely determines the crop’s latitudinal adaptation and yield potential [15]. Under the long-day and low-temperature conditions prevalent in high-latitude regions, the *PhyA-E1-FT* pathway plays a dominant role in regulating soybean flowering, with temperature further modifying these flowering mechanisms [16,17]. For instance, the *PhyA-E1* pathway is pivotal for flowering at 35 °C, while at 30 °C *GmFT2a* and *GmFT5a* are upregulated through *E1*-independent pathways to promote flowering and maturity [18].

The *E*-series genes (*E1–E4*) play a fundamental role in determining soybean cultivar distribution across latitudes, collectively accounting for 62–66% of maturity variation [19]. *E1*, a soybean-specific transcription factor, is particularly influential, as single-site variations can dramatically alter geographic distribution [20]. The molecular pathway involves *GmPHYA3* (*E3*) and *GmPHYA2* (*E4*) acting as photoreceptor modules that perceive long-day signals and transmit them through multiple pathways, including *E2*, *GmPRR3a/GmPRR3b-GmLHYs*, and *GmELF3* (*J*)*-GmLUXs*. These signals induce the expression of *E1* and its homologs (*E1La* and *E1Lb*), ultimately regulating downstream flowering genes [21].

The *FT* gene family serves as a crucial downstream target in this regulatory network. The core regulatory factors, *E1s*, target four major downstream genes: *GmFT2a/GmFT5a*, *GmFT4*, and *GmFT1a*. Through mechanisms involving *GmFUL2a*, *E1s* downregulate the flowering inducers *GmFT2a* and *GmFT5a*, which then directly or indirectly *GmSOC1a*, influence flowering time [22]. Notably, *GmFT2a* and *GmFT5a* exhibit coordinated regulation that enables soybean adaptation across diverse photoperiods, while *GmFT1a* acts as a flowering suppressor with an opposite expression pattern, contributing to adaptation under both long and short photoperiods [23].

Recent research has identified additional regulatory genes enhancing high-latitude adaptation. Mutations in *GmPRR3b*, particularly when combined with *GmPRR3a*, accelerate flowering and maturity [24]. *GmSOC1a* influences both flowering time and yield components, with *GmSOC1a ^G^* emerging as the predominant haplotype in high-latitude regions [25]. Similarly, the two alleles of *GmFUL2a*, *GmFUL2a-H1* and *GmFUL2a-H2*, have undergone both natural and artificial selection to improve high-latitude adaptability [13].

Nonetheless, prior research has primarily concentrated on soybeans in mid-latitude and southern low-latitude areas, while the genetic foundation of maturity of soybean cultivars from locations north of 50° N remains largely unexamined [11,26,27,28]. This study seeks to delineate the phenotypic traits of soybean maturity in areas north of 50° N and to examine the genotypes of 35 significant genes associated with maturity. These genes cover various key nodes of soybean photothermal regulation pathways. The primary objective is to establish a theoretical basis for improving the photothermal adaptation of soybean cultivars in high-latitude locations and to promote the extension of soybean cultivation to elevated latitudes and altitudes.

## 2. Results

### 2.1. Precise Classification of Test Soybean Cultivars into Relative Maturity Groups (RMGs)

A total of 438 soybean varieties were grown in Heihe for two years (HH2020 and HH2021), and the developmental stages of the soybeans, such as emergence (VE), the beginning bloom (R1), and physiological maturity (R7), were recorded. To verify the stability of the experimental data, we conducted a fitting analysis on the days from VE to R7 over the two years. The results of a linear regression significance test (*F*-test) revealed significant relationships between HH2020 and HH2021 (*p* < 0.01). The *R*^2^ values of the regression models exceeded 0.8, indicating a strong fit of the regression lines to the data (Figure 1a). To further refine the RMG classification, the phenotypic mean values of the MG reference cultivars in HH2020 and HH2021 were used to generate regression equations correlating with the VE-R7 period with RMGs (Figure 1b).

The RMGs of the cultivars included in this study ranged from RMG −2.2 (MG 0000.8, negative values to represent maturity groups adapted to high-latitude, short-growing-season regions) to RMG 0.5 (MG 0.5) (Appendix A). Among them, nine cultivars had an RMG ≤ −2.1. The most common RMG group was RMG −0.1 (MG 00.9) to RMG −1.0 (MG 000.0), which included 186 cultivars, followed by RMG −1.1 (MG 000.9) to RMG −2.0 (MG 0000.0) with 159 cultivars, and RMG 0.0 (MG 0.0) to RMG 0.5 (MG 0.5) with 84 cultivars.

### 2.2. Phenotypic Analysis in Multiple Environments

The experimental sites Heihe (HH), Beijicun (BJC), and Labudalin (LBDL) exhibit distinct latitudinal and altitudinal differences, influencing light and temperature conditions. BJC, located 3 degrees of latitude north of HH, has an altitude 141 m higher, resulting in 0.5 h longer average daylength than HH, and the average temperature is about 3 °C lower than HH. LBDL, situated at a similar latitude to HH but at an altitude 423 m higher, experiences a lower average temperature (3 °C lower than HH) during the period from emergence to the beginning of maturity (Appendix A). These environmental variations contribute to differences in light and temperature conditions across the three locations.

The phenotypic analysis of matured soybean cultivars from high-latitude regions revealed significant diversity in their maturity (Figure 1c). A notable observation was that most cultivars were immature before the first frost in BJC and LBDL (Figure 1d,f,g,i). Significant differences in maturity were observed between locations HH and LBDL (*p* < 0.01), suggesting that varying temperature substantially influences the maturity of high-latitude soybean cultivars (Figure 1c). Additionally, significant differences in maturity were noted between HH and BJC, further highlighting the combined effects of photoperiod and temperature interactions on maturity (Figure 1c).

In 2020, we identified 438, 152, and 150 cultivars that matured before the first frost in HH, BJC and LBDL, respectively (Figure 1d–f). In 2021, the same number of cultivars matured in HH and BJC, while 145 cultivars matured before the first frost in LBDL (Figure 1g–i). The RMG ranges for cultivars that matured before the first frost were −2.2 to 0.5 in HH, −2.2 to −0.9 in BJC, and −2.2 to −1.1 in LBDL (Appendix A).

Based on maturity and yield considerations, we recommend introducing soybean cultivars with RMG values of 0.0, −1.0, and −1.2 into regions with similar photoperiod and temperature conditions as those in HH, BJC, and LBDL.

### 2.3. Photothermal and Temperature Sensitivity of Soybean in High-Latitude and High-Altitude Regions

To investigate whether the sensitivity of soybean cultivars to daylength and temperature changes during the expansion to high-latitude and high-altitude areas, this study calculated the temperature response sensitivity and photothermal comprehensive response sensitivity of the soybean cultivars. The results showed that there were significant differences in temperature response sensitivity and photothermal comprehensive response sensitivity among the soybean cultivars (Appendix A).

The RMG values were negatively correlated with temperature sensitivity (Figure 2a). However, no significant differences were found in temperature response sensitivity across different maturity groups (Figure 2b). In contrast, a significant variation in photothermal comprehensive response sensitivity was observed among the cultivars, with a positive correlation with RMG values (Figure 2c). The difference between the MG 00 and MG 000 was extremely significant, while there was no significant difference in the sensitivity of the comprehensive photothermal response between other adjacent maturity groups (Figure 2d). In conclusion, as the photoperiod shortens, soybean cultivars tend to become more sensitive to temperature fluctuations, while their sensitivity to long-day conditions decreases. This trend suggests an adaptation mechanism in which soybeans optimize their response to long-day, low-temperature conditions, facilitating their expansion to high-latitude and high-altitude regions.

### 2.4. Genetic Variation Analysis and Haplotype Identification of Maturity Major-Effect Genes in Soybean

Whole-genome resequencing was conducted on 438 soybean cultivars using the Illumina NovaSeq platform, achieving an average depth of 10X clean reads per cultivar. The clean reads were aligned to the soybean reference genome (*Glycine max* Wm82.a4. v1), resulting in an average mapping coverage of 93% (Appendix A). A total of 11,388,299 single-nucleotide polymorphisms (SNPs) were detected across all cultivars. Following rigorous quality control filtering, 3,926,223 high-quality SNPs were retained, distributed across all 20 soybean chromosomes. Notably, the majority of these SNPs were concentrated on chromosomes 4, 15, 18, and 19 (Appendix A).

Based on the Wm82.a4.v1 reference genome and previous research [29], among the 35 major flowering and maturity genes tested, most exhibited varying degrees of mutation. For instance, upstream genes *E1–E4* in the photoperiod pathway underwent loss-of-function mutations, which may be essential for soybean adaptation to the long-day, low-temperature environment of high-latitude regions. In contrast, genes such as *GmFT1a* and *GmFT2a*, which, located downstream, were more conserved, maintained their functional integrity [29]. Additionally, 23 genes displayed the same variation across the tested cultivars, forming a shared genetic basis for soybean adaptability in high-latitude regions (Appendix A).

These mutations were evaluated by comparing the coding regions. From these, 12 genes were selected for further analysis, resulting in the identification of 19 SNPs and four InDels within the coding regions of these genes (Appendix A). Among these polymorphisms, a single nucleotide substitution at position 16145 (A > T) in *E2* and at position 5203 (C > T) in *E3* resulted in premature termination codons (Figure 3a,j). Furthermore, a deletion of a single base (A) at positions 3726 and 4099 in *E4* caused a frameshift mutation (Figure 3b), while a single base insertion (T) at position 2649 in E3 also resulted in a frameshift mutation (Figure 3a). The remaining variations consisted of missense mutations and synonymous mutations (Figure 3).

To investigate the genetic basis of soybean adaptation to long-day, low-temperature environments, haplotypes were identified for twelve major-effect maturity genes using exon variation data from the 438 resequenced soybean cultivars (Figure 4). Association analysis was performed between these haplotypes and maturity traits across six environments, with trials conducted at three sites over two years. BLUE (Best Linear Unbiased Estimator) values were calculated using the R package “lme4”, followed by association analysis to examine the relationship between the identified haplotypes and soybean maturity (Figure 5).

By comparing the maturity of different gene haplotypes, we found that natural variations in *GmPHYA3 (E3)*, *GmPHYA2 (E4)*, *GmGBP*, *GmELF3 (J)*, *GmFUL2a*, *GmLHY1a*, *GmFT2b*, *GmFT5b*, and *GmLHY2a* were significantly correlated with soybean maturity distribution (Figure 5).

The polymorphic loci in the *E3* exon region were classified into four haplotypes. *E3^H2^* contains a frameshift mutation, while *E3^H4^* carries a premature stop codon (Figure 4a). As a result, the maturity of the *E3^H1/3^* cultivars was significantly later compared with the *E3^H2/4^* cultivars (Figure 5a). Similarly, for the *E4* gene, three haplotypes were identified, with *E4^H2^* and *E4^H3^* both carrying frameshift mutations (Figure 4b). The maturity of the *E4^H1^* cultivars was significantly later compared with the *E4^H2/3^* cultivars (Figure 5b).

In *GmELF3* (*J*), two haplotypes resulting in amino acid substitutions (Figure 4c) were associated with a significantly later maturity of the *GmELF3^H2/4/8^* cultivars compared with the *GmELF3^H1/3/5/6/7^* cultivars (Figure 5c). *GmLHY2a* was categorized into two haplotypes based on a single SNP (Figure 4d). Cultivars with the *GmLHY2a^H1^* haplotype matured significantly earlier than those carrying the *GmLHY2a^H2^* haplotype, which contains a missense mutation (Figure 5d). In *GmLHY1a*, a combination of one SNP and one InDel resulted in the classification of the gene into four haplotypes (Figure 4e). Cultivars with the *GmLHY1a^H3^* haplotype showed significantly earlier maturity compared with those with *GmLHY1a^H1/2/4^* haplotypes (Figure 5e).

Although no amino acid changes were detected among the three SNPs in *GmFT2b* (Figure 4f), the maturity of the *GmFT2b^H2^* cultivars was significantly later compared with the *GmFT2b^H1/3/4/5^* cultivars (Figure 5f), possibly due to promoter region variations affecting gene expression. In *GmFT5b*, two haplotypes were identified based on a single SNP (Figure 4g), with the *GmFT5b^H^*^2^ haplotype, containing a missense mutation, leading to earlier maturity compared with the *GmFT5b^H1^* haplotype (Figure 5g).

For *GmFUL2a*, a missense mutation (Cys > Ser) gave rise to two haplotypes (Figure 4h), with cultivars carrying the *GmFUL2a^H1^* haplotype maturing significantly earlier than those with *GmFUL2a^H2^* (Figure 5h).

Lastly, three haplotypes were defined for *GmGBP* (Figure 4i), where the maturity of the *GmGBP^H2^* cultivars were significantly later compared with the *GmGBP^H1/3^* cultivars (Figure 5i).

Out of the 35 major genes associated with the soybean growth period, 22 genes displayed no amino acid functional variation and were found to be functionally conserved across the tested cultivars. Notably, the *E1* gene consistently exhibited the *e1-as* type mutation in all test varieties, which is likely essential for soybean adaptation to long-day, low-temperature environments (Table 1A). In contrast, the remaining 12 genes exhibited considerable natural variation among the test cultivars. Among these, no significant differences in growth period were observed between different haplotypes of the *E2*, *GmFLC-like,* and *GmSOC1a* genes. However, significant variations in growth period were observed between the haplotypes of the other nine genes (Table 1B; Figure 4). These findings indicate that early-maturing haplotypes of these nine genes, along with their specific combinations, play a pivotal role in enabling soybean adaptation to environments with higher latitudes and altitudes, characterized by distinct light and temperature conditions.

### 2.5. Distribution Pattern of Different Haplotypes of Maturity Groups and Geographical Regions

To evaluate the impact of major-effect maturity genes on the distribution of maturity groups of soybean cultivars in high-latitude regions, we analyzed the haplotype distribution frequencies of nine genes with significant haplotype differences among the 438 soybean cultivars. The results showed that the distribution frequency of early-maturing haplotypes of nine genes in the MG 000-0000 cultivar was significantly higher compared with the MG 0-00 cultivar (Figure 6).

We further examined the haplotype distribution patterns of these nine major maturity genes in adaptive soybean cultivars in three experimental locations. The results indicated that, compared with HH, the distribution frequency of *E3^H2/4^*, *E4^H2/3^*, *GmGBP^H1/3^*, *GmELF3^H1/3/5/6/7^*, *GmFUL2a^H2^*, *GmLHY1a^H3^*, *GmFT2b^H1^*, *GmFT5b^H2^*, and *GmLHY2a^H1^* was significantly higher in both BJC and LBDL (Figure 7). This pattern aligns with the observed maturity phenotype, indicating that these genes play a crucial role in regulating the maturity of soybean cultivars in high-latitude regions.

## 3. Discussion

### 3.1. Characteristics of Adaptive Soybean Cultivars for High Latitudes and High Altitudes

Soybean originated in the middle latitudes of China and gradually expanded to the higher latitudes due to global warming and developments of breeding techniques [30]. During this expansion, different ecological types adapted to varying latitudes were developed. In northern Northeast China, soybean cultivars suitable for planting were typically early-maturing, extremely early-maturing, or super-early-maturing types, with maturity groups (MGs) ranging from MG 0 to 0000 [7,9].

In this study, HH was reconfirmed as the northernmost region in China suitable for planting MG 0 soybean cultivars [31]. To further explore cultivar adaptation, BJC, located at a higher latitude, and LBDL, situated at a higher altitude but at the same latitude as HH, were selected as additional test sites. A set of soybean cultivars suitable for planting in different regions was screened. For practical production purposes, the cultivars chosen for BJC, HH, and LBDL corresponded to RMG −1.0 (MG 00.0), RMG 0.0 (MG 0.0), and RMG −1.2 (MG 000.8), respectively. These studies provide a basis for introducing MG 000-00 cultivars for planting in Huma, Tahe, and Mohe in Heilongjiang province, as well as MG 000 cultivars for planting in Genhe and Yakeshi in Inner Mongolia.

As latitude or altitude increased, the temperature sensitivity of soybean cultivars also increased, leading to a shortening of their maturity periods. However, no significant differences were observed between the maturity groups. The photothermal response sensitivity decreased as the maturity shortened (Figure 2), indicating that cultivars adapted to high-latitude or high-altitude regions formed a unique type of photothermal response. These cultivars were relatively insensitive to photoperiods but more sensitive to temperature.

### 3.2. Conservation and Variation Patterns of Major-Effect Maturity Genes in Soybean Varieties North of 50° N Latitude

Natural variation and the combination of genes during the growth stages are important genetic bases of different ecological types [11,12,13,14,27,30,32,33]. The proportion of recessive variation at key loci, such as *E1–E4*, which are associated with high-latitude cultivars, increases as the latitude of the cultivars increases [19,26]. The haplotype composition of FT family genes and clock genes also changes with latitude distribution [11,34].

The results of this study showed that the *e1* locus was of the *e1-as* type across all tested cultivars [19], and there were no mutations in the flowering suppressor gene *GmPRR3b* or the flowering promoter genes *GmFT2a/5a* [11,19,27]. These results suggest that allelic variation of *E1* and these highly conserved genes may be essential for local soybean cultivars to adapt to long-day and low-temperature environments. Haplotype analysis showed that the phenotypes of *E2*, *GmFLC-like*, and *GmSOC1a* did not vary significantly across different haplotype growth stages. This insensitivity to photoperiod and temperature suggests that these genes play a minimal role in the maturity variation among tested cultivars. These loci, showing the same variation and no phenotypic differences, form the common genetic basis for adaptation to alpine ecological conditions.

In this study, *E3^H1^* corresponds to *e3* in previous studies, *E3^H2^* corresponds to *e3*-*fs*, *E3^H4^* is *e3*-*ns*, and *E3^H3^* is a new haplotype [19]. *E4^H1^* corresponds to *e4* in previous studies, and *E4^H2/3^* corresponds to *e4-kes* in previous studies [19]. The recessive mutation of the *E1* site, that is, the formation of the ecological type, adapted to the low altitude of 50° N latitude (MG 0-MG 00). On the basis of recessive mutations in *E1*, *E3*, and *E4* loci of the cultivars adapted to higher altitudes or higher latitudes (MG 000-MG 0000), the precocity haplotypes at *GmGBP1*, *GmFUL2a*, *GmFT2b*, and *GmFT5b* loci accumulated was the main genetic basis for the growth stage diversity of soybean cultivars in the high-latitude cold region [11,12,13,14].

Flowering regulatory genes play an important role in the process of soybean adaptation to high latitudes. As more genes are discovered and identified, the soybean photoperiod flowering regulatory network is becoming increasingly sophisticated [11,13,20,23]. Based on the soybean photoperiod core regulatory network *E3/E4-J-E1-FT* and the results of haplotype analysis, this study proposes a dominant haplotype combination and molecular regulatory network of key genes for photothermal response in soybean cultivars in regions north of 50° N latitude (Appendix A).

Flowering regulatory genes play important roles in the process of soybean adaptation to high latitudes. *E3* and *E4*, as soybean photoreceptors [35,36], sense the long-day signal in high-latitude regions and transmit it to downstream circadian clock systems, activating the expression of the flowering-promoting genes *GmLHY1a^H3/2aH1^* and *GmELF3^H1/3/5/6/7^* [37], and upregulating the transcription levels of the flowering-promoting genes *GmGBP^H1/3^*, *GmFUL2a^H1^*, and *GmFT2b^H1/3/4/5^*/*5b^H2^* by inhibiting the transcriptional activity of *E1*, ultimately promoting soybean flowering and maturation [13,16,38,39,40].

Flowering regulation pathways in soybean exhibit distinct characteristics compared with model plants such as *Arabidopsis* and rice [41]. Recent studies have identified a unique *phyA-LUX-E1-FT* photoperiodic flowering pathway in soybean, which differs fundamentally from the *phyB-CO-FT* pathway predominant in many non-leguminous species [17]. Notably, the phyA-mediated photoperiodic flowering mechanism is not exclusive to soybean, as evidenced by its conserved regulatory role in other leguminous species including medicago (*Medicago truncatula*) and pea (*Pisum sativum*) [42,43]. *E1* is a specific gene in the gene regulatory network of flowering in soybean. Phylogenetic analyses of short-day (SD) leguminous crops reveal functional conservation of *E1* between soybean and common bean (*Phaseolus vulgaris*), where it suppresses flowering through transcriptional repression of *GmFT2a/GmFT5a* or *PvFT* homologs. On the other hand, in the long-day plant Medicago, the function of *MtE1L* is shifted to flowering promoting [44]. Flowering regulation in leguminous plants demonstrates relative evolutionary conservation, suggesting that our analyses of flowering genes may provide foundational insights for other legumes.

### 3.3. Molecular Breeding Strategies for Improving Super-Early Soybean Cultivars in High-Latitude Regions

At present, the primary strategy for breeding soybean cultivars suitable for regions north of latitude 50° N involves promoting recessive variations in the *E1* locus and aggregating early-maturity haplotypes such as *E3^H2/4^*, *E4^H2/3^*, *GmGBP^H1/3^*, *GmELF3^H1/3/5/6/7^*, *GmFUL2a^H2^*, *GmLHY1a^H3^*, *GmFT2b^H1^*, *GmFT5b^H2^*, and *GmLHY2a^H1^* [32]. This strategy results in an ecological type adapted to higher latitudes and altitudes, characterized by longer sunlight durations and lower temperatures.

Recently, genome editing, especially the clustered regularly interspaced short palindromic repeats/CRISPR associated protein (CRISPR-Cas) technology, has been widely applied in crop breeding [45,46]. The CRISPR-Cas technology is also widely used for the improvement of soybean agronomic traits [47,48]. Based on our results, it is feasible to employ the CRISPR-Cas technology to induce mutations in maturity-related genes such as *E1*–*E4*, thus converting Heihe 43, a soybean variety widely planted in China, into an earlier-maturing one. These improved varieties are expected to be planted in high-latitude areas like the northern part of Northeast China. Ultimately, the integration of genome-wide association studies (GWAS) with genomic selection (GS) can expedite the identification of advantageous variants for early maturity. Utilizing high-throughput sequencing data, breeders can identify quantitative trait loci (QTLs) that regulate photoperiod and temperature responses, thereby enhancing marker-assisted selection (MAS). Furthermore, transcriptome and epigenetic analyses will yield enhanced understanding of the regulatory processes governing flowering time adaptability. The integration of these methodologies with precision phenotyping and environmental modeling will improve breeding efficiency and facilitate the creation of soybean cultivars that exhibit enhanced adaptation to high-latitude and high-altitude areas.

Our present study, which focuses on the natural variations in key maturity genes of super-early soybean cultivars, has established a solid foundation for the molecular breeding of, and improving the adaptability of, soybean to extra high-latitude regions. However, there remains substantial scope for further exploration. Future research should prioritize collaborative endeavors. For example, conducting comprehensive gene expression studies, and integrating advanced climate modeling and simulation analysis are crucial steps. These efforts will not only validate and build upon our current findings but also significantly deepen our understanding of the intricate mechanisms governing soybean adaptation in the unique environment combing long days and low temperature.

## 4. Materials and Methods

### 4.1. Plant Materials and Multiple-Site Field Experiments

A total of 438 soybean cultivars were collected from high-latitude regions worldwide, including 327 from Northeast China, 49 from Far-Eastern Russia, 40 from North America, and 22 from Sweden.

The 438 cultivars were grown under natural day length conditions in BJC (53°28′ N, 122°21′ E, altitude 295 m), HH (50°15′ N, 127°28′ E, altitude 154 m), and LBDL (50°15′ N, 120°19′ E, altitude 577 m) in 2020 and 2021 in China to investigate the maturity dates. Maturity in the HH location was used for precise RMG classification [28]. The cultivars were manually sown in two rows (2 m in length, with a plant spacing of 5 cm and row spacing of 60 cm) on May 9th of each year. After emergence (VE), the plants were thinned to maintain 30 uniform, healthy individuals per row. Flowering time was recorded at the R1 stage, which is defined as the point when there is at least one open flower at any node on the main stem. Maturity was recorded at the R7 stage, which is defined as the point when at least one normal pod on the main stem has reached its mature pod color [49].

Each location–year combination was considered an experimental environment, named BJC2020, BJC2021, HH2020, HH2021, LBDL2020, and LBDL2021, respectively. These cultivars represent the maturity group MG0000-0, as shown in Appendix A.

### 4.2. Photothermal Sensitivity Calculation


(1)
$$TS(\%)=\frac{DTB_{LBDL}-DTB_{HH}}{DTB_{LBDL}}\times 100\%\;$$


The formulas for calculating temperature sensitivity (TS), represented by Function (1) and photothermal comprehensive response sensitivity (PTCRS), represented by Function (2), are as follows. *DTB_LBDL_* represents the days from emergence to beginning of bloom under LBDL temperature conditions, and *DTB_HH_* represents the days from emergence to beginning of bloom under HH temperature conditions.(2)$$PTCRS(\%)=\frac{DTB_{BJC}-DTB_{HH}}{DTB_{BJC}}\times 100\%$$

Similarly, *DTB_BJC_* represents the days from emergence to beginning of bloom under BJC temperature conditions, and *DTB_HH_* represents the days from emergence to beginning of bloom under HH temperature conditions.

### 4.3. DNA Extraction, Resequencing, SNP Calling, and Natural Variation Analysis

Genomic DNA was extracted from each sample using the Plant Genomic DNA Kit (TIANGEN, Beijing, China). Genome sequencing libraries were constructed using the DNA Library Prep Kit (Vazyme, Nanjing, China). The libraries were sequenced on the Illumina NovaSeq platform. Raw reads were quality-trimmed using Trimmomatic (ILLUMINACLIP:adaptor.sequence:2:30:10, TRAILING:3, LEADING:3, SLIDINGWINDOW:4:10, MINLEN:20, TRAILING:3) [50]. Clean reads were then mapped to the soybean reference genome (*Glycine max* Wm82.a4.v1) using BWA-mem with the default parameters [51]. SNP calling and annotation were performed with SAMtools 1.21 [52], BCFtools 1.21 [53], and SnpEff 4.3 [54]. Subsequently, only high-quality SNPs (max-missing ≤ 0.1, quality score (QUAL) ≥ 50.0, depth (DP) ≥ 5.0, quality by depth (QD) ≥ 5.0, mapping quality (MQ) ≥ 30, coverage ≥ 90, max/min-alleles = 2, minor allele frequency (MAF) ≥ 0.01) were retained using VCFtools v0.1.16 [55]. Finally, BCFtools was used to download variation information for natural variation analysis. In addition to identifying SNPs, the resequenced data were analyzed to detect functional mutations in the coding regions of 35 major flowering and maturity genes. Specific attention was given to identifying loss-of-function mutations in *E1*, *E2*, E3, and *E4*, which may be crucial for soybean adaptation to high-latitude environments. Gene variants, including single nucleotide polymorphisms (SNPs), insertions (InDels), and frameshift mutations, were further categorized. These variations were linked to significant phenotypic differences in growth stages and maturation across the cultivars.

### 4.4. Haplotype Analysis

Haplotype analysis of SNPs and InDels in the exons of twelve major-effect maturity genes was performed using Python programs v3.10. The Best Linear Unbiased Estimator (BLUE) values of eight environments were calculated with the “lme4” package [56]. The stacked column chart of MGs was generated using the “ggplot2” package.

### 4.5. Statistical Analysis

The RMG for all cultivars in the HH location was calculated using a linear regression model, with the period from VE to R7 as the dependent variable, based on the assumed RMG [57]. Reference cultivars for each MG were used to calculate the VE-R7 regression. To compare the mean values for each measured parameter, one-way analysis of variance (ANOVA) or two-sample Student’s *t*-tests were conducted, as appropriate, using SPSS 21.

## Figures and Tables

**Figure 1 ijms-26-03362-f001:**
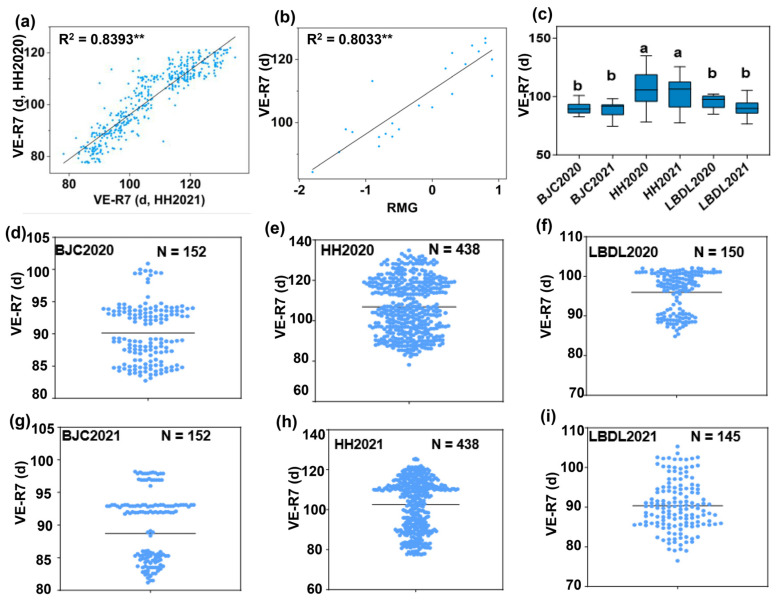
Influence of different environments on soybean maturity. (**a**) Fitting analysis of the days from emergence (VE) to physiological maturity (R7) of the tested varieties over two years in Heihe (HH). (**b**) Linear regression equation between the relative maturity group and the days to physiological maturity of standard soybean varieties within the maturity group. (**c**) Comparison of the days to physiological maturity of the tested soybean varieties in different years and at different locations. (**d**–**f**) Varieties that could mature in Beijicun (BJC) (**d**), HH (**e**), and Labudalin (LBDL) (**f**) in 2020 and the distribution of their days to physiological maturity. (**g**–**i**) Varieties that could mature in BJC (**g**), HH (**h**), and LBDL (**i**) in 2021 and the distribution of their days to physiological maturity. “BJC2020” and “BJC2021” represent Beijicun (BJC) in 2020 and 2021, respectively; “HH2020” and “HH2021” denote Heihe (HH) in 2020 and 2021, respectively; “LBDL2020” and “LBDL2021” correspond to Labudalin (LBDL) in 2020 and 2021, respectively. “d” indicates days; VE-R7 means the days from emergence (VE) to physiological maturity (R7); “N” represents the number of matured cultivars; “RMG” refers to relative maturity group. “**” in (**a**,**b**) mean close fitting of data for two years. Significant differences are shown by different letters following Duncan’s multiple-range test (*p* < 0.05).

**Figure 2 ijms-26-03362-f002:**
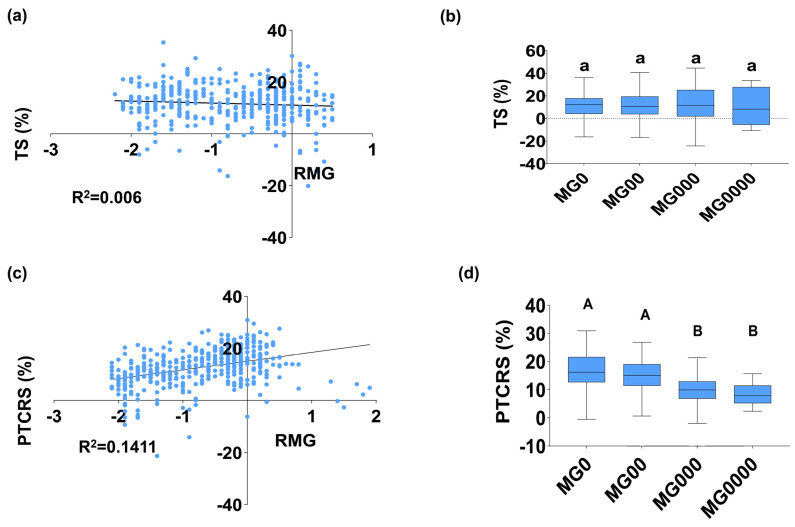
Analysis of TS and PTCRS in the tested varieties. (**a**) Correlation analysis of the temperature sensitivity (TS) and the relative maturity group (RMG) of 438 tested varieties. (**b**) Comparison analysis of the TS of 438 tested varieties in different maturity groups. (**c**) Correlation analysis of the photothermal comprehensive response sensitivity (PTCRS) and the relative maturity group (RMG) of 438 tested varieties. (**d**) Comparison analysis of the PTCRS of 438 tested varieties in different maturity groups. Significant differences are shown by different letters following Duncan’s multiple-range test (*p* < 0.05).

**Figure 3 ijms-26-03362-f003:**
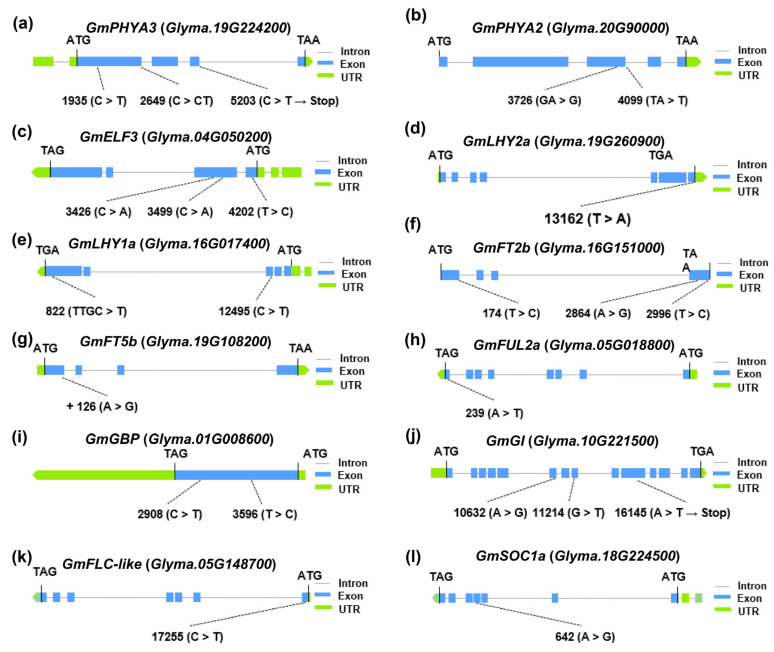
Variations in the coding sequences of 12 major-effect maturity genes. (**a**) *GmPHYA3*, (**b**) *GmPHYA2*, (**c**) *GmELF3*, (**d**) *GmLHY2a*, (**e**) *GmLHY1a*, (**f**) *GmFT2b*, (**g**) *GmFT5b*, (**h**) *GmFUL2a*, (**i**) *GmGBP*, (**j**) *GmGI*, (**k**) *GmFLC-like*, and (**l**) *GmSOC1a*. “Stop” indicates a premature termination codon.

**Figure 4 ijms-26-03362-f004:**
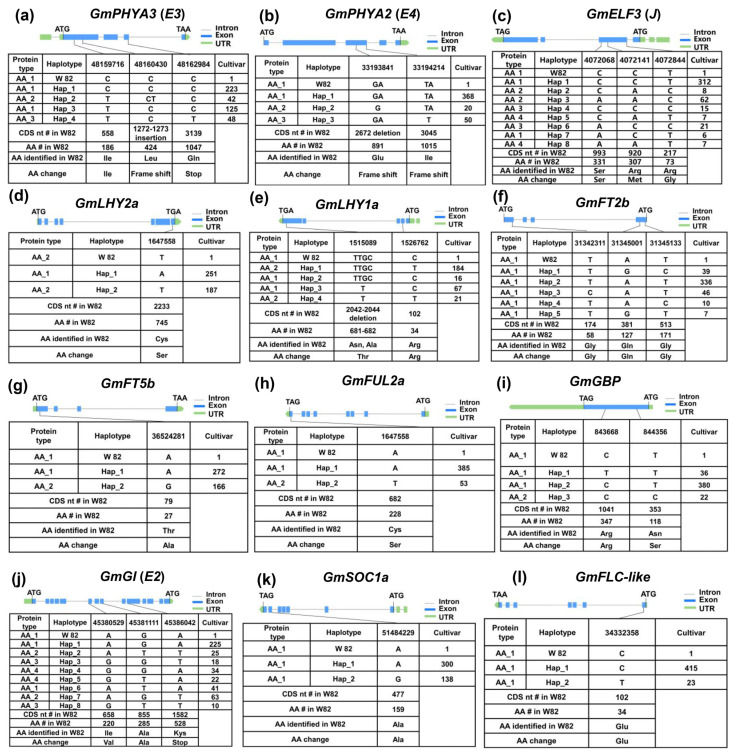
Haplotypes and amino acid (AA) sequences of 12 major-effect maturity genes in the 438 resequenced soybean cultivars. (**a**) *GmPHYA3*, (**b**) *GmPHYA2*, (**c**) *GmELF3*, (**d**) *GmLHY2a*, (**e**) *GmLHY1a*, (**f**) *GmFT2b*, (**g**) *GmFT5b*, (**h**) *GmFUL2a*, (**i**) *GmGBP*, (**j**) *GmGI*, (**k**) *GmFLC-like*, and (**l**) *GmSOC1a*. “#” means the position of CDS nt and AA. “Stop” indicates a premature termination codon; W82 denotes Williams 82; Hap means haplotype.

**Figure 5 ijms-26-03362-f005:**
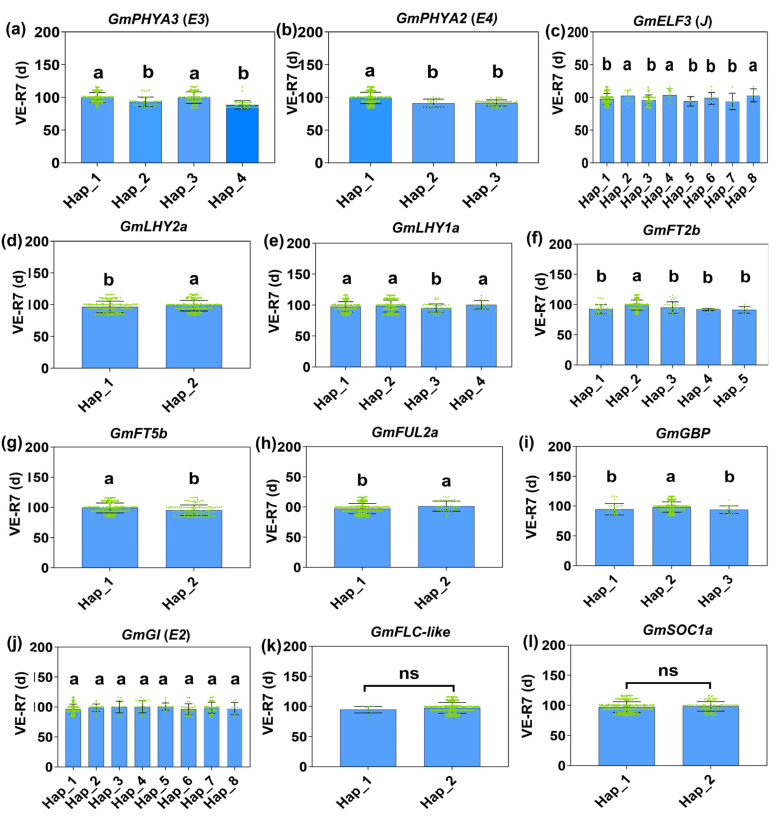
Haplotype distribution patterns of 12 major-effect maturity genes in super-early soybean cultivars adapting to high-latitude regions. (**a**) *GmPHYA3*, (**b**) *GmPHYA2*, (**c**) *GmELF3*, (**d**) *GmLHY2a*, (**e**) *GmLHY1a*, (**f**) *GmFT2b*, (**g**) *GmFT5b*, (**h**) *GmFUL2a*, (**i**) *GmGBP*, (**j**) *GmGI*, (**k**) *GmFLC-like*, and (**l**) *GmSOC1a*. The green dots represent the varieties contained in each haplotype. “d” indicates day; “ns” denotes no significant difference. Significant differences are shown by different letters following Duncan’s multiple-range test (*p* < 0.05).

**Figure 6 ijms-26-03362-f006:**
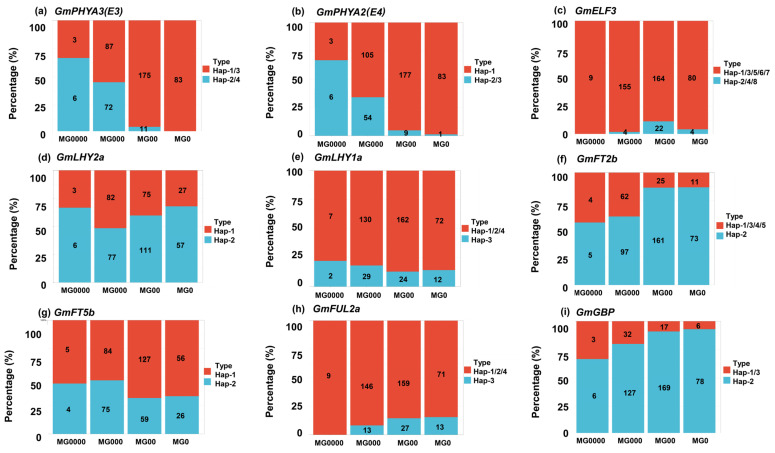
Distribution patterns of nine major-effect maturity genes across maturity groups in different soybean cultivars. (**a**) *GmPHYA3*, (**b**) *GmPHYA2*, (**c**) *GmELF3*, (**d**) *GmLHY2a*, (**e**) *GmLHY1a*, (**f**) *GmFT2b*, (**g**) *GmFT5b*, (**h**) *GmFUL2a*, and (**i**) *GmGBP*. Hap means haplotype.

**Figure 7 ijms-26-03362-f007:**
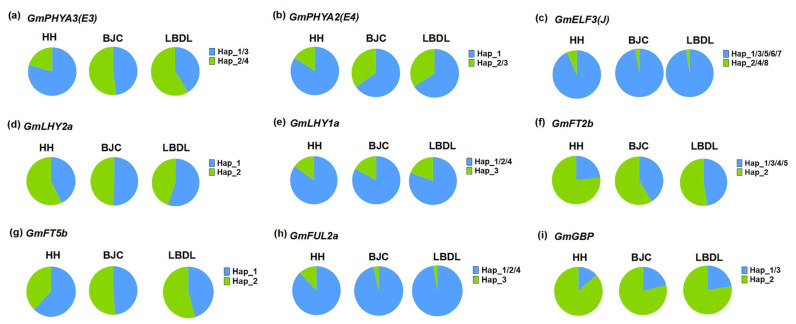
Distribution patterns of different haplotypes of 9 major-effect maturity genes across 3 experimental locations. (**a**) *GmPHYA3*, (**b**) *GmPHYA2*, (**c**) *GmELF3*, (**d**) *GmLHY2a*, (**e**) *GmLHY1a*, (**f**) *GmFT2b*, (**g**) *GmFT5b*, (**h**) *GmFUL2a*, and (**i**) *GmGBP*. Hap means haplotype; HH refers to Heihe; BJC refers to Beijicun; LBDL refers to Labudalin.

**Table 1 ijms-26-03362-t001:** Classification of maturity-related genes in soybean cultivars adaptive for high latitudes beyond 50° N.

**A. Genes Functionally Conserved Across the Cultivars**		
**Gene**	**Status**	**Type**
*E1La*	Functional	Inhibitory
*E1Lb*	Functional	Inhibitory
*GmFT1a*	Functional	Inhibitory
*GmFT1b*	Functional	Inhibitory
*GmFT2a*	Functional	Promotive
*GmFT3a*	Functional	Promotive
*GmFT5a*	Functional	Promotive
*GmFT4*	Functional	Inhibitory
*GmLHY1b*	Functional	Promotive
*GmLHY2b*	Functional	Promotive
*GmLUX1*	Functional	Promotive
*GmLUX2*	Functional	Promotive
*GmPRR3a*	Functional	Inhibitory
*GmPRR3b*	Functional	Inhibitory
*GmFULa*	Functional	Promotive
*GmSOC1b*	Functional	Promotive
*QNE1*	Functional	Promotive
*GmPIF4b*	Functional	Promotive
*GmRAV*	Functional	Inhibitory
*GmEID1*	Functional	Promotive
*GmTFLc*	Functional	Inhibitory
*GmTFLd*	Functional	Inhibitory
*E1*	Mutation (e1-as)	Inhibitory
**B. Genes Specifically Mutated in the Cultivars Adaptive for High Latitudes Beyond 50° N**		
**Gene**	**Haplotype**	**Function**
*GmPHYA3(E3)*	H2/H4	Early maturity
*GmPHYA2(E4)*	H2/H3	Early maturity
*GmELF3*	H1/H3/H5/H6/H7	Early maturity
*GmLHY2a*	H1	Early maturity
*GmLHY1a*	H3	Early maturity
*GmFT2b*	H1/H3/H4/H5	Early maturity
*GmFT5b*	H2	Early maturity
*GmFUL2a*	H2	Early maturity
*GmGBP*	H1/H3	Early maturity
*E2*	H1-H8	No significant
*GmFLC-like*	H1/H2	No significant
*GmSOC1a*	H1/H2	No significant

## Data Availability

Data is contained within the article and Appendix A.

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
