# Peer review of "Natural Variations in Key Maturity Genes Underpin Soybean Cultivars Adaptation Beyond 50° N in Northeast China"

_ijms, 2025, doi:10.3390/ijms26073362_

Round 1
Reviewer 1 Report
Comments and Suggestions for Authors
This study investigated the genetic basis of soybean maturity in the northern region of China, as much has been said about this in other areas in China. The introduction, result and discussion are well-written. Specified below are some additional methods that could be supplemented to improve the manuscript further.
- Gene expression analysis is required to elucidate the transcriptional regulation of key maturity genes and their responses to photothermal conditions.
- Integrating climate modelling and simulation analysis could help predict the impact of climate change on soybean cultivation and adaptation in the northern regions.
- Phylogenetic analysis could provide insights into the evolutionary history of soybean maturity genes and their relationships with other leguminous species.
Moreover, the references should be rearranged.
Reviewer 2 Report
Comments and Suggestions for Authors
Expanding soybean planting area is very important for food security in China. While 50° N 16
latitude marks the northern boundary of major soybean regions, improving adaptability to photothermal conditions enables expansion to higher latitudes and altitudes. Understanding the genetic basis of super-early maturity of soybean is crucial to achieving this goal. In this manuscript, 438 soybean germplasms collected from high-latitude regions were evaluated in three different latitude areas and produced significant results. In general, the manuscript was well-written and experimental design was suitable and I have the following comments:
- After I downed the manuscript, I did not see any figures, thus, it is difficult for me to make the decision? ,
- Minor comment, the first letter of each key words should be capital;
- There is no references cited from 2025, reference need to be upated.
Reviewer 3 Report
Comments and Suggestions for Authors
The work presented here is a comprehensive study of the genetic basis of adaptation of soybean cultivars to photothermal conditions at latitudes above 50o N. The authors conducted an in-depth analysis of natural variation and haplotypes in 35 key genes that regulate flowering and maturation, which contributes to our knowledge of the genetic mechanisms underlying soybean adaptation to the short growing season and low temperatures.
The paper was developed by a team of authors representing five different research institutions, which positively contributed to the interdisciplinary nature of the research and the broad coverage of the problem. The set research goal - identification of genetic variants for adaptation of soybeans to high latitudes - has been achieved, and the results can form the basis for further research into the improvement of phenological traits of soybeans and their breeding adaptation to the climatic conditions of regions located in northern China and the Far East of Russia.
Contributions to science include:
1. Comprehensive genetic analysis - the use of resequencing data to identify SNPs and InDels in maturity-related genes enabled precise characterization of genetic drivers of adaptation.
2 Accuracy in identifying key mutations - the authors showed that the e1-as allele in the E1 gene was present in all cultivars tested, which is likely a key factor for adaptation to long-day, low-temperature conditions. In addition, nonsense mutations in E2 and E3 and frameshift mutations in E3 and E4 were identified.
3. haplotype analysis - Conducting an analysis of haplotypes and their impact on soybean phenology allowed us to better understand the processes of accumulation of favorable genetic variants in different growing regions.
Areas potentially to be completed in further research work:
1. statistical analysis - the paper lacks more complex statistical analyses that would synthesize the results obtained. It is worth considering the use of multivariate analysis of variance (ANOVA) with interactions, which could provide additional information on the effect of environment on the expression of maturation genes.
2 Influence of environmental conditions - an important uncontrolled factor in the conducted experiment was temperature, in particular the occurrence of frosts before or at the beginning of soybean maturation. The authors rightly took into account the effect of the influence of different environments in the 2020 and 2021 seasons (Fig. 1d-i), an important addition is a more detailed representation of the course of weather conditions (Fig. S1).
3 Further exploration of the figures - the collected genotypic and phenotypic material has great potential for further analysis, especially in the context of identifying additional sub-populations within the study material.
Technical notes:
- Fig. 2d - uppercase letters are used in the designations of homogeneous groups, while lowercase letters are used in the earlier graphs. It is recommended to standardize the format.
- Fig. 4 - in the printed version the graph is illegible, while the electronic version is not objectionable. It is worth improving the readability of the graphic in the printed version.
In conclusion, the article is a valuable contribution to the study of soybean adaptation to high latitudes and provides a solid foundation for further breeding research.
Round 2
Reviewer 1 Report
Comments and Suggestions for Authors
The authors have addressed the comments raised. The authors claimed expression analysis and other comments raised are limitations of the study and will be addressed in the future. They further indicated this study's limitation in the manuscript, allowing further studies. As such, I propose the manuscript can be accepted and published in its present form.
